# The core metabolome and root exudation dynamics of three phylogenetically distinct plant species

Sarah McLaughlin[1,3], Kateryna Zhalnina [1], Suzanne Kosina [1], Trent R. Northen [1,2] ✉ & Joelle Sasse [1,2,3] ✉

Root exudates are plant-derived, exported metabolites likely shaping root-associated microbiomes by acting as nutrients and signals. However, root exudation dynamics are unclear and thus also, if changes in exudation are reflected in changes in microbiome structure. Here, we assess commonalities and differences between exudates of different plant species, diurnal exudation dynamics, as well as the accompanying methodological aspects of exudate sampling. We find that exudates should be collected for hours rather than days as many metabolite abundances saturate over time. Plant growth in sterile, nonsterile, or sugar-supplemented environments significantly alters exudate profiles. A comparison of *Arabidopsis thaliana*, *Brachypodium distachyon*, and *Medicago truncatula* shoot, root, and root exudate metabolite profiles reveals clear differences between these species, but also a core metabolome for tissues and exudates. Exudate profiles also exhibit a diurnal signature. These findings add to the methodological and conceptual groundwork for future exudate studies to improve understanding of plant-microbe interactions.

Plants roots exude a tremendous variety of compounds into the rhizosphere. Exudates influence carbon and nutrient cycles[1–3], shape physiochemical properties of soils, and act as nutrients and signals to root-associated microbial communities[4,5]. Root exudates were suggested as an important functional trait for plant ecology and studies, as they correlated with plant nutrient-use strategies[6].

Root exudates are discussed as one factor on how plant-microbiome interactions are shaped. Evidence for this comes from many observations that were made in microbiome as well as in exudate studies: Microbiomes as well as exudate profiles are distinct between plant species, cultivars, wild and modern varieties[5,7–12]. Microbiome composition and exudation changes with plant developmental stage, diurnal timepoint, and with abiotic and biotic stresses[5,13–19]. For example, sugar exudation decreases and organic acid exudation increases along a developmental gradient in several monocot and dicot plant species[15,20,21]. Also, diurnal fluctuations were observed for

some exuded compounds such as some lipids[3], single organic acids such as citrate[22] and mugineic acid[23], and flavonoids and glucosinolates[24]. Single studies link changes in microbiomes with altered exudation: in Arabidopsis, altered triterpene exudation altered microbiome composition[25], malic acid exudation was altered by a leaf pathogen infection, attracting a belowground biocontrol microbe[18], and exuded phenolic acids were metabolized selectively by plant-associated microbes[4].

A range of methods was developed for exudate analysis. Plants can be grown in sterile or nonsterile settings, in soil or hydroponic culture, and exudates can be collected in a variety of solutions such as deionized water, nutrient medium, or soil washes. Most experimental factors alter the plants environment and thus have the potential to change plant metabolism and with it, exudation. For single factors, such an impact on exudation was shown. For example, exudates collected in deionized water compared to a growth medium often

[1]Lawrence Berkeley National Laboratory, Environmental Genomics and Systems Biology, Berkeley, CA, USA. [2]DOE Joint Genome Institute, Lawrence Berkeley National Laboratory, Berkeley, CA, USA. [3]Present address: Institute for Plant and Microbial Biology, University of Zurich, Zurich, Switzerland. ✉e-mail: trnorthen@lbl.gov; jschlaepfer@botinst.uzh.ch

contained higher levels of carbon and amino acids[15,26,27], likely due to the strong osmotic imbalance between root and solvent.

Exudate collection duration also influences the metabolic profile. The rate of total organic carbon exudation (carbohydrates and organic acids) was estimated higher when rice exudates were collected for 2 h than for 4 h or 6 h, independent of the plant developmental stage[15]. External amino acid concentrations of a variety of plant species increased in the first few hours of exudation up to one day and remained stable afterwards[15,28]. Another major determinant for detection of exudates is the sterile or nonsterile growth environment. For tomato grown in nonsterile conditions, no sugars or organic acids could be measured in exudates, likely due to microbial activity, whereas sterile tomato exhibited organic acid exudation in the μM range within hours[29]. In nonsterile setups, sterilizing agents can be added to block microbial metabolism, but they likely alter the plant exudation profile as well[27]. However, the dynamics of exudation of most chemical classes remains unexplored.

Few studies analyzed exudates from soil-grown plants, which is usually done with spatial resolution using membranes or leachate collection, or by washing pots with water or organic solvent[30,31]. The presence of microbes in nonsterile settings or of a soil matrix leads to the detection of a complex metabolite profile. The identification of exudate metabolites in such experimental setups is not trivial. To summarize, multiple studies showed that exudation is a dynamic process, depending on a number of environmental and experimental factors.

Although it becomes clear that exudation is a dynamic process, most parameters shaping exudation have not been systematically explored. Most studies focus on how a few experimental or biological parameters change exudation of one or few chemical classes. Comprehensive studies analyzing the dynamics of a large number of exuded metabolites systematically for the influence of experimental factors such as duration of exudate collection or impact of growth medium on exudation are missing, as well as a comprehensive analysis of similarities and differences in exudates of different plant species.

Here, we show that both, experimental as well as biological factors impact root exudation. We systematically assessed two technical and two biological aspects: (i) the influence of the collection duration on the exudate profile, (ii) the influence of the growth environment (sterile, nonsterile, sugar supplemented) on the exudation profile, (iii) the similarities and differences between exudate and tissue metabolic profiles of multiple phylogenetically distant plant species and (iv) the diurnal signature associated with metabolic profiles of plant tissues and exudates. We find that a collection window of a few hours is best suited to capture the dynamics of many exuded compounds, but a window of a few days might be necessary for low-abundant compounds. Further, higher levels of carbohydrates are detected in exudates of sucrose-supplemented conditions compared to control conditions. Exudation profiles and root morphology significantly change between sterile and nonsterile conditions. About two thirds of compounds of roots and exudates of dicot *Arabidopsis thaliana*, the monocot *Brachypodium distachyon*, and the legume *Medicago truncatula* are detected in all three species. The remaining one third of compounds is specific to one or two species. Further, we find a diurnal signature for 7−32% of exuded metabolites (depending on the species). With this publication, we introduce the concept of the core metabolome: The core metabolome comprises metabolites present in most plant species, in analogy to the core microbiome associated with plant tissues. We hypothesize that the core exudates are responsible for recruitment of the core microbiome to roots. In contrast, the exudates specific to a species or plant family result in more specific plant-microbe associations.

## Results

### Most metabolites are detected with short exudate collection times, but longer collection times increase signal

We first assessed the technical questions to determine the exudation collection parameters for the remaining experiments presented here. In a first experiment, we aimed at determining the temporal dynamics of exuded metabolites to determine the optimal exudate collection duration, the shortest timepoint of exudate collection still allowing for a good signal, and possible saturation of exudation signals at later timepoints. For this, a model monocot (*B. distachyon*), a model dicot (*A. thaliana*) and a model legume (*M. truncatula*) were grown for three weeks in a sterile hydroponic setup (Supplementary Fig. 1). The growth medium was replaced at the beginning of the collection period, and exudates were collected in the growth medium for five different durations, ranging from 0.5 h to 4 d. For the three plant species, 63 exudate metabolites were detected overall (metabolite identification with exact mass, retention time, and fragmentation spectrum matching our in-house library, Supplementary Data 1). Among the identified compounds were amino acids and other organic acids, carbohydrates, nucleosides, nucleotides and derivatives, and benzenoids. A temporal dynamic was found for 16% (*M. truncatula*), 37% (*B. distachyon*), and 52% (*A. thaliana*) of compounds in any pairwise comparison, with the other metabolites remaining at similar levels throughout the experiment (Supplementary Figs. 2 and 3). Principal component analysis (PCA) showed a continuous shift for exudate profiles collected over time for two of the three species. *M. truncatula* displayed more variation overall, and a less clear separation of the time points, possibly to the higher genetic diversity of the seedstock used for this species (Fig. 1). Most differences were found between the earliest and latest timepoint (0.5 h and 4 d, Supplementary Fig. 2). An in-depth analysis of temporal patterns of *B. distachyon* metabolites distinct between these timepoints revealed a variety of temporal behaviors, from linear, exponential or logarithmic increases to bell-shaped dynamics (Supplementary Fig. 3). Most compound intensities increased over time. Decreases were only observed in few instances between 1 d and 4 d. Further, all metabolite classes investigated showed similar temporal dynamics (Supplementary Fig. 3, Chi square test). The choice of a collection time thus does not bias the detection towards specific chemical classes.

To choose the optimal exudate collection duration, the metabolite abundance data was converted to presence-absence (higher than background in 50% of more of samples), allowing the comparison of metabolite numbers present at certain timepoints. With the latest timepoint (4 d) set to 100%, we detect 16% of compounds at 0.5 h, 32% at 2 h, 61% at 4 h, and 84% at 1 d. We conclude that for experiments that are not time sensitive such as comparisons of ecotypes or mutants, longer collection times maximize compound number and intensity. However, long collection times may obscure differences present at earlier timepoints, and factors such as metabolite re-uptake are likely more relevant at higher metabolite concentrations, complicating the picture[28,32,33]. For experiments with high temporal resolution, we recommend a collection window of a few hours. For all further experiments presented here, a 2 h exudate collection window was chosen.

### Plant growth conditions influence exudate profile

Historically, plants for exudate collection are grown in three broad categories of environments: sterile, nonsterile, or sugar-supplemented growth media (aside from soil-grown plants, which are not considered here). Sterile environments allow a focus solely on plant-derived compounds, whereas nonsterile environments include microbe-derived compounds but allow exudate collection of large plants. Sucrose-supplemented conditions are often used in *A. thaliana* research to improve growth and reduce phenotype heterogeneity, but likely alter the plant's carbohydrate metabolism[4,24,34,35]. We assessed if

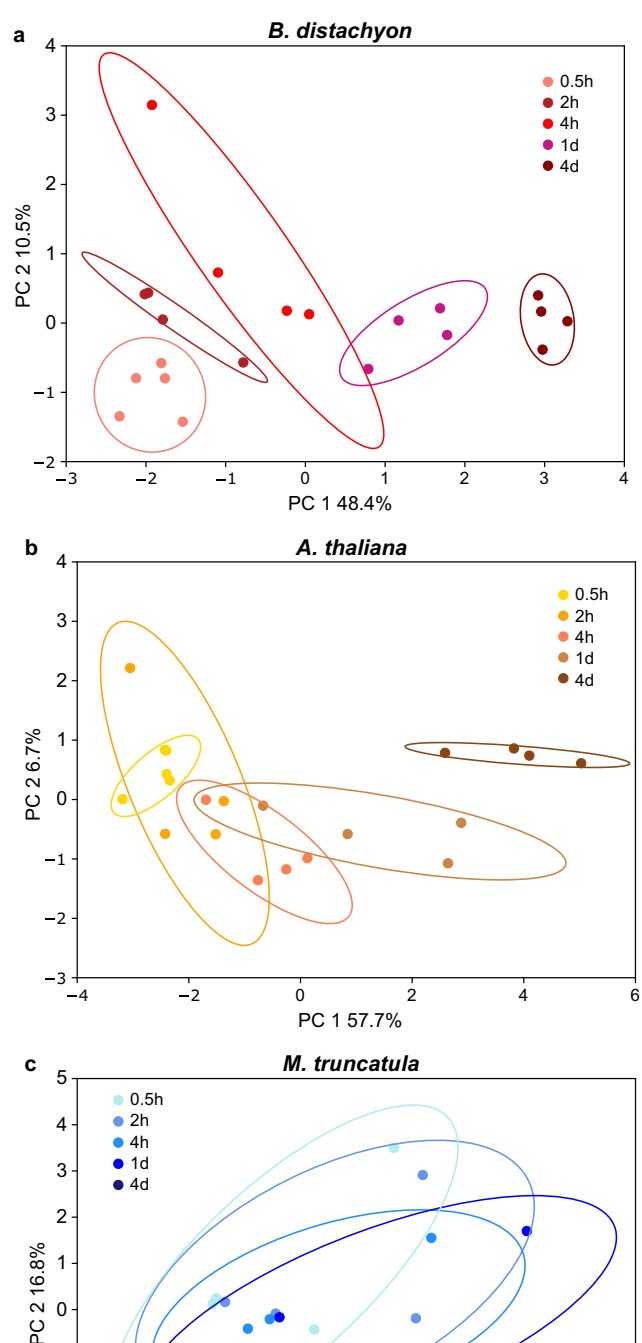

**Fig. 1 | Root exudation dynamics in three species.** Principal component plots of root exudates collected after 0.5 h, 2 h, 4 h, 1 d, 4 d of *B. distachyon* **a**, *A. thaliana* **b**, and *M. truncatula* **c**. Details of this dataset are given in Supplementary Figs. 2 and 3. Total number of metabolites: 63. Number of jars with 3–5 plants: 3–4 for each timepoint. One representative experiment out of three total is displayed.

root exudation is distinct in these conditions by growing *B. distachyon* in sterile, nonsterile, and sugar-supplemented environments. A phenotypic analysis revealed increased shoot and reduced root weight of nonsterile-grown plants compared to sterile-grown plants, resulting in

a lower root/shoot ratio (Fig. 2a, b). Nonsterile-grown plants also formed more lateral, crown, and tertiary roots, resulting in approximately twice as many roots in total (Supplementary Fig. 4). Tissue weights and root morphology of sucrose-grown plants was comparable to sterile-grown plants.

Exudates of plants grown in these environments were collected for 2 h in sterile medium for all plants, to avoid an influence of medium composition on the downstream analytical pipeline. All growth conditions resulted in clearly distinct exudation profiles on a PCA plot (Fig. 2c). Of the 109 compounds identified, 37% were significantly different between sterile and sucrose-supplemented conditions, 53% between sterile and nonsterile and 68% between nonsterile and sucrose-supplemented conditions (Anova/Tukey test, $p < 0.05$). Overall, 78% of compounds are distinct in at least one pairwise comparison.

Sucrose-supplemented plants exuded more carbohydrates than sterile-grown plants (and exhibited a trend for increased amino acid exudation), and they exuded less organoheterocyclic compounds and organic acids (Fig. 2d). Compared to sterile conditions, sucrose-supplemented plants exclusively exuded a C6 sugar alcohol, a dihexose, glutamic acid and its methylated form, and methylglutarate (first five compounds in Fig. 3a), among others. Glutamic acid serves as a hub in amino acid metabolism, as it can be converted to glutamine and ornithine, and its conversion to 2-oxoglutarate is needed for many enzymatic conversions of other amino acids, such as aspartic acid. Many of the amino acids connected to glutamic acid via a metabolic pathway are also upregulated in sucrose-supplemented plants. Glutamine, glutamic acid and aspartic acid metabolism also influences nucleoside and nucleotide metabolism, and some nucleosides are also exuded at higher levels in sucrose-supplemented conditions (Fig. 3a). Although plants are autotrophic organisms, they seemingly grow mixotrophically here, take up supplemented sugar, convert it via carbohydrate metabolism, which impacts in turn amino acid and nucleoside metabolism and exudation.

Nonsterile-grown plants show trends for higher levels of benzenoids and lower exudation of carbohydrates as well as nucleosides and derivatives (Fig. 2d). Generally, the single compounds affected are distinct from the ones changing in sucrose-supplemented conditions. Several amino acids are present at higher levels compared sterile conditions. Also, many organic acids (e.g. fumaric acid), benzenoids (e.g. benzoic acid and derivatives), and organoheterocyclic compounds (e.g. quinolinecarboxylic acid and derivatives) are higher in nonsterile than in sterile conditions (Fig. 3b). Many of these compounds have aromatic rings. Interestingly, aromatic organic acids are metabolized by plant-associated bacteria, in contrast to soil-associated bacteria[4]. Thus, many of these compounds might play a role in plant-microbe interactions. All compounds with increased presence in nonsterile conditions were also detected in exudates collected in sterile conditions, although sometimes in much smaller amounts (Supplementary Data 1). Thus, the increase in metabolite levels in nonsterile conditions might be due to an alteration in plant metabolism, maybe in response to microbial presence, or due to the additional presence of microbial metabolism. Compounds reported to be microbe-specific were not detected in this dataset. Interestingly, few compounds were decreased in abundance in nonsterile conditions compared to sterile environments. As microbes are heterotrophs and as they must utilize plant-derived compounds for growth, there are two potential explanations: either the compounds depleted by microbes were not part of this dataset, or the exudation rate of the compounds of interest increased simultaneously as the consumption by microbes, making the depletion.

We conclude that different growth conditions do result in clearly distinct exudation profiles. Comparisons of experiments performed in different growth conditions are therefore not trivial and should be avoided. All further data presented here is from plants grown in sterile, non-sucrose supplemented conditions.

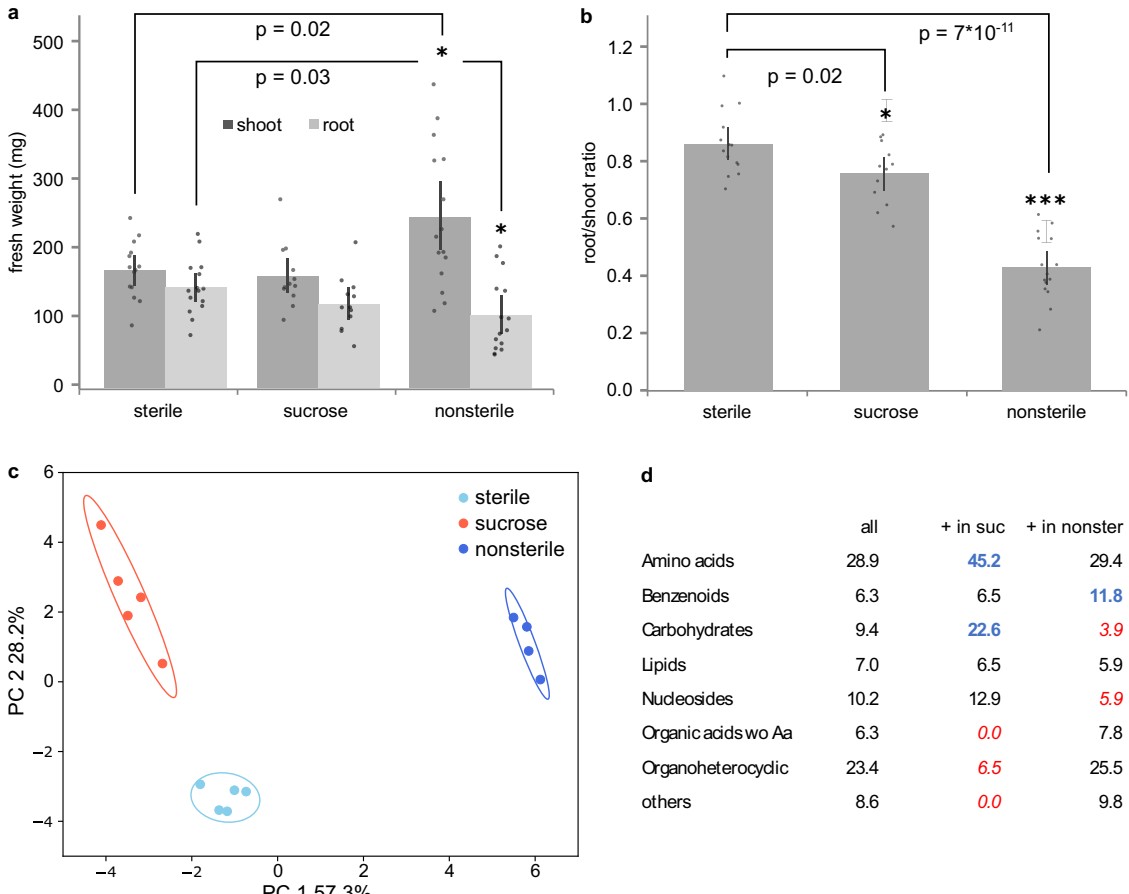

**Fig. 2 | *B. distachyon* root exudation dynamics in different growth conditions.** *B. distachyon* was grown for 3 weeks in sterile, nonsterile, and sucrose-supplemented conditions followed by exudate collection in the same 0.5 strength MS medium. Fresh weight of root and shoot **a** and root/shoot ratio **b** at 3 weeks. Data are averages ± SEM, *n* = 15 individual plants, *=*p* < 0.05, ***=*p* < 0.0005 (two-sided t-test). Root morphology of plants is described in Supplementary Fig. 4. **c** Principal component analysis of exudate profiles. **d** Metabolite classes detected in the entire dataset, and classes enriched (bold, blue) or depleted (italic, red) in sucrose-supplemented or nonsterile conditions compared to sterile conditions (Fishers exact test). Values are in percent of total number of metabolites (*n* = 109 metabolites of 4–5 jars with 3 plants per growth condition). One representative experiment out of three is displayed.

## Distinct root and shoot metabolic profiles of three plant species

Distinct plant species have different metabolic capabilities and assemble distinct microbiomes. We wondered to what degree their exudate profiles differ, and if there was a diurnal signature detectable in exudates. *B. distachyon*, *A. thaliana*, and *M. truncatula* were chosen as representatives of monocot, dicot, and dicot legumes for diurnal exudate collection. Root and shoot tissues were also collected at end of day and end of night.

Overall, 143 metabolites were identified in roots and shoots of the three plant species. In shoot tissues, 85% of the detected compounds were significantly different in at least one pairwise comparison, whereas in root tissues, only 38% of compounds were distinct. Both tissue types however separated clearly according to plant species on PCA plots (Supplementary Fig. 5a–f). Largest differences were detected between the monocot *B. distachyon* shoot compared with dicot shoots (67%), whereas the differences between the dicot shoots was about half, comparatively (39%, Supplementary Fig. 5g). Generally, differences between roots were smaller than between shoots, with *B. distachyon* vs. *M. truncatula* being the largest with 27%, and the other two comparisons smaller with 20% (Supplementary Fig. 5h). A heatmap of significantly distinct root metabolites revealed presence of six clusters (Fig. 4). The two clusters 2 and 4 comprised of metabolites with highest abundance in a single species, *B. distachyon*, whereas the other clusters comprised of metabolites high in two species compared to another with lower abundant metabolites: cluster 1, 6: *A. thaliana*

(A), *B. distachyon* (B) > *M. truncatula* (M), cluster 3: M, B > A, cluster 5: A, M > B. In shoots, many metabolites are of high abundance in *B. distachyon* compared to the other species, followed by *M. truncatula* and *A. thaliana* (Supplementary Fig. 6). When comparing roots versus shoots in a single species, clearly distinct metabolic profiles were found also. Largest differences were found in *M. truncatula* with 76%, followed by *A. thaliana* with 33%, and by *B. distachyon* with 19% (Supplementary Fig. 5I). No diurnal signature could be detected in tissue data when plotted by tissue or by species (Supplementary Fig. 5b–f).

## Distinct exudate profiles of three plant species with a diurnal signature

Metabolic profiles of root exudates were also clearly distinct between the three different species (Fig. 5a–c). Although no diurnal differences were detected in tissues, exudates displayed a diurnal signature (medium changed at beginning of day and night, respectively, exudates collected for 2 h, Fig. 5a, Supplementary Fig. 7a–c). A total of 74 metabolites was evaluated in this experiment. *A. thaliana* exhibited most differences between exudate samples collected at the end-of-day vs. those collected at the end-of-night, and *M. truncatula* the least (Fig. 5d). Overall, 77% of exuded metabolites differed between two species at the end-of-day, and only 28% in any end-of-night comparison. Both timepoints exhibited most differences between *A. thaliana* and *B. distachyon*, followed by *A. thaliana* and *M. truncatula* (Fig. 5b, c).

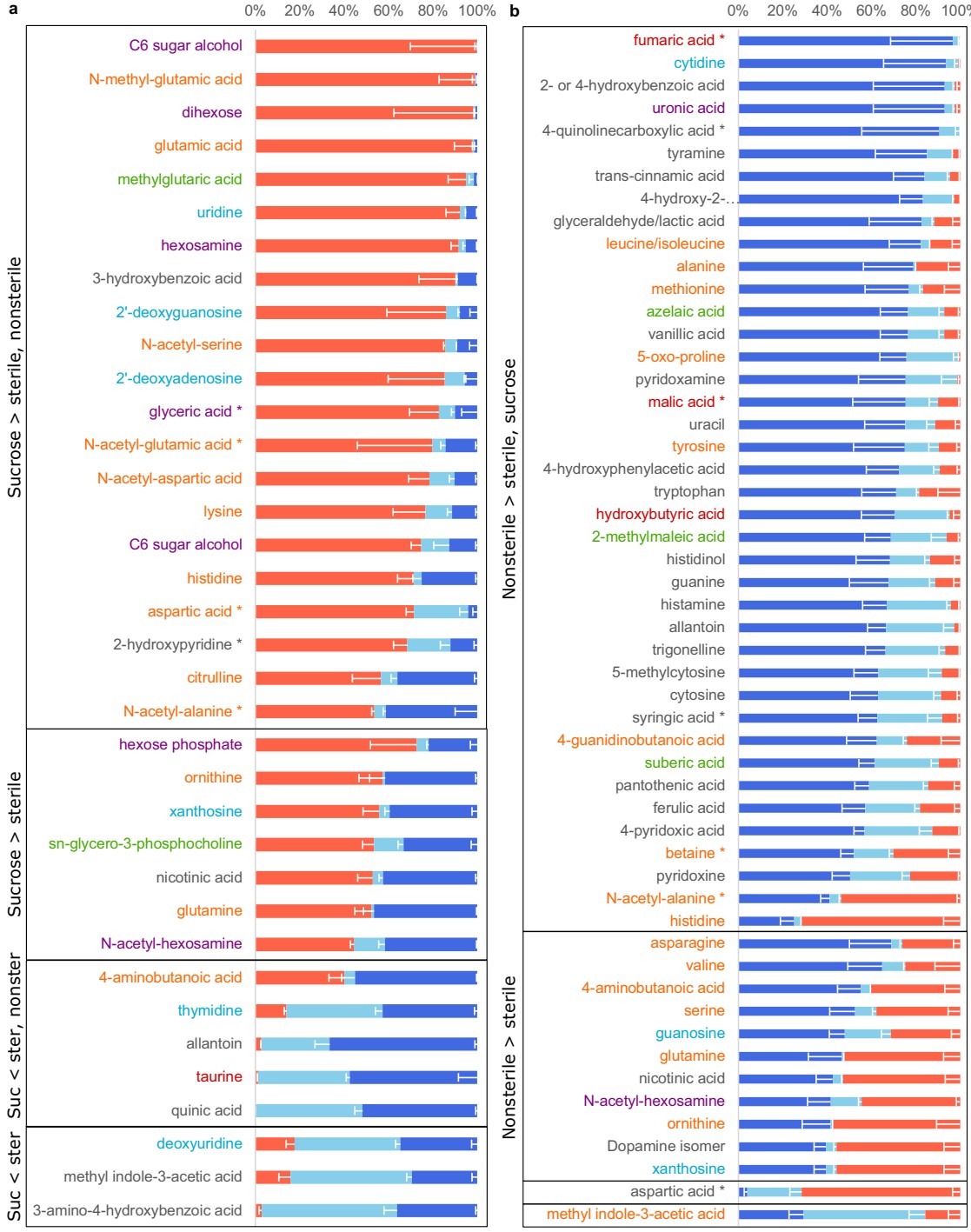

**Fig. 3 | Metabolites distinct in *B. distachyon* exudates from different growth conditions.** Normalized peak height of metabolites significantly different between growth conditions (light blue: sterile, red: sucrose supplemented, dark blue: nonsterile)(Anova/Tukey test, *p* < 0.05, *p*-values are given in Supplementary Data 1). Sucrose-supplemented vs sterile conditions **a**, nonsterile vs sterile conditions **b**. Displayed are all conditions as reference, black boxes and labeling to the left indicate in which conditions a metabolite is distinct, and if it its abundance is increased or decreased compared to control. S: Sterile, NS: nonsterile, 4-hydroxy-2: 4-hydroxy-2-quinolinecarboxylic acid, *: and/or isomers. Data are averages ± SEM, *n* = 4–5 jars per growth condition with 3 plants each. Metabolites are colored by class: orange: Amino acids, peptides and derivatives, purple: carbohydrates and conjugates, green: lipids and lipid-like, blue: nucleosides, nucleotides and derivatives, red: organic acids, black: other.

Within one species, 48% of metabolites differed between end-of-day and end-of-night. Taken together, interspecies differences were largest in shoots, followed by end-of-day exudates, roots, and end-of-night exudates (Fig. 5b, c, Supplementary Fig. 5g, h).

Over 90% of compounds distinct between the two diurnal time-points were lower at the end-of-night compared to end-of-day, only

cinnamic acid, asparagine, and agmatine sulfuric acid increased (Fig. 5e, value > 1, dotted line). Diurnally distinct metabolites belonged to various chemical classes. A Fishers exact test revealed more changes than expected by chance for organic acids, lipids, and less changes than expected for nucleosides (Supplementary Fig. 7d). Significant changes in exudate diurnal patterns were rarely consistent across

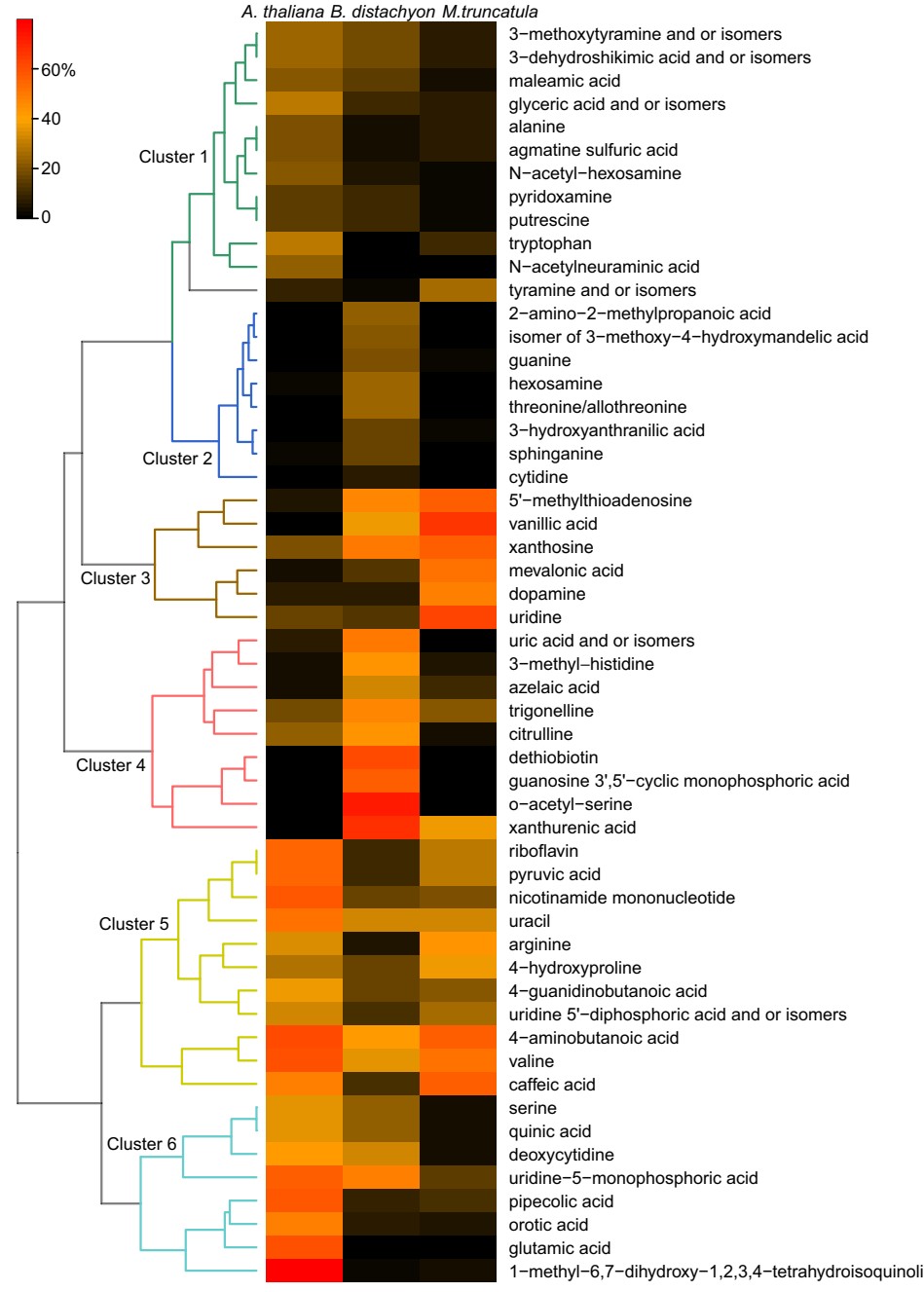

**Fig. 4 | Species-specific patterns of root metabolites.** Heatmap of root metabolites. Each cell represents an averaged value of normalized peak height of metabolites significantly different between plant species (Anova, *p* < 0.05 and post-hoc Duncan's multiple range test). Hierarchical clustering identified six clusters of metabolites based on their distribution across analyzed species. Heatmap of shoot metabolites is found in Supplementary Fig. 6, and PCA plots of tissues in Supplementary Fig. 5.

species (Fig. 5). Overall, we conclude that exudates show a diurnally distinct metabolic profile, with most compounds being of lower abundance at the end of night than at the end of day consistent with lower rates of exudation at night.

## The core metabolome of roots and exudates

We were further interested which compounds were present in roots and exudates of all three plant species, as they could represent a 'core metabolome'. These compounds could be present in roots and exudates of many plant species and could thus be important for interaction with the core microbiome associated with plant roots.

To investigate this question, the relative abundance data of metabolites was converted to presence/absence data (present if above

background in more than 50% of samples), as the metabolite extraction procedures of exudates and tissues did not allow for quantitative comparison. Overall, 150 metabolites were detected in roots and exudates. Inspection revealed that half of the metabolites (43%) was present in roots of all species, comprising the root core metabolome (Fig. 6). One fifth of metabolites (21%) was detected in exudates of all species, comprising the core exudate metabolome (Fig. 6). The exudate core metabolites were also detected in roots of all species and are thus a subset of the core root metabolome (total root core metabolome: 43% + 21% = 64% of all metabolites). The chemical classes detected in the root and exudate core metabolomes were similar, comprising of amino acids and other organic acids, nucleosides and derivatives, carbohydrates and conjugates, and in the case of roots,

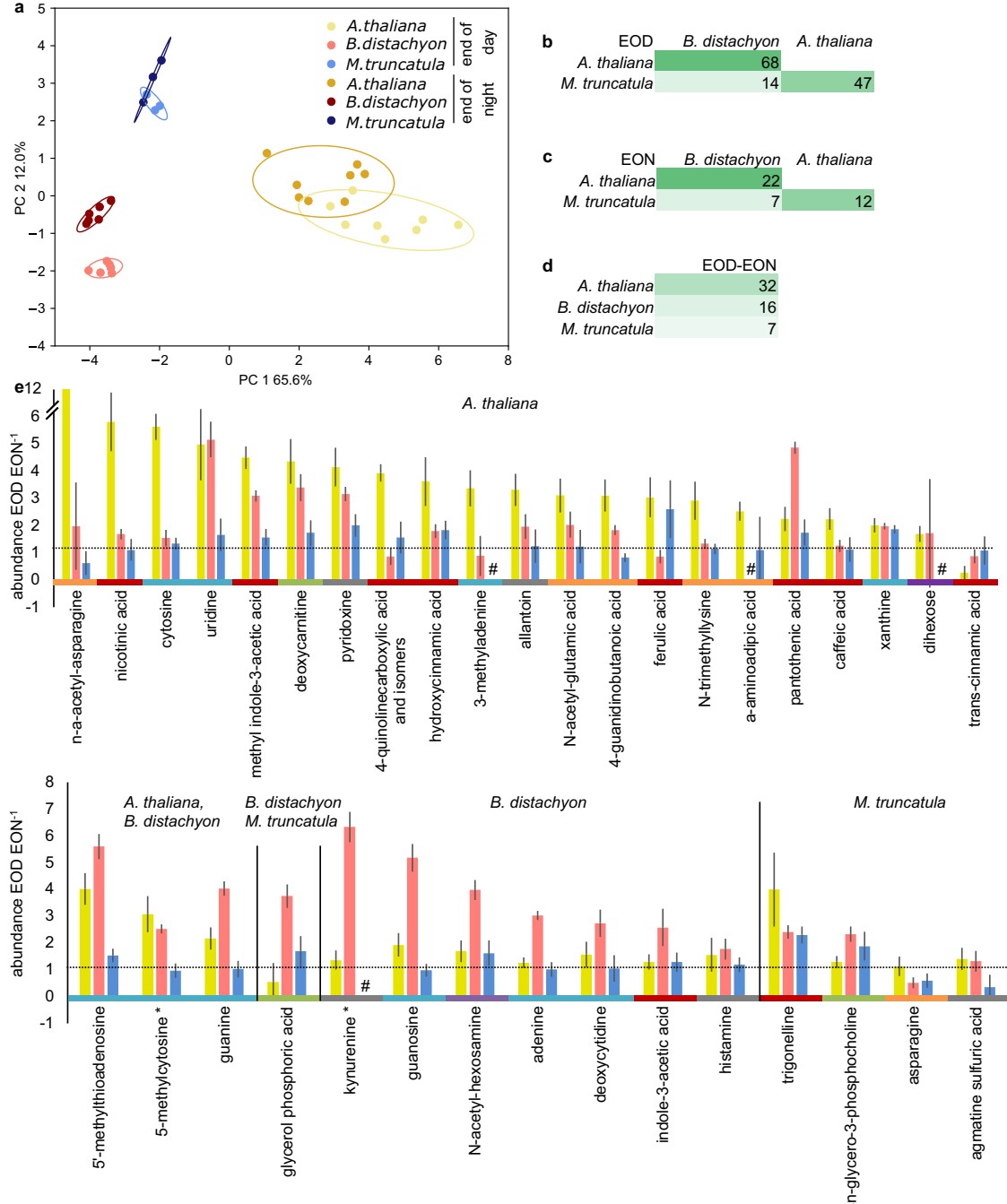

**Fig. 5 | Diurnal changes in exudate profiles. a** Principal component analysis of *A. thaliana* (yellows), *B. distachyon* (reds) and *M. truncatula* (blues) exudate metabolites at the end of day (EOD) and end of night (EON). Percent of metabolites that are significantly different between species at end of day **b**, end of night **c**, and for each species between end of day and night **d**. Colored by high (dark green) to low (light green) intensity. **e** Fold difference of metabolites significantly increased in end of day vs. end of night *A. thaliana* (yellow, top graph), in *B. distachyon* (red), *M. truncatula* (blue), or in two species (bottom graph, indicated in subtitles). One representative experiment out of three total is displayed. Compounds are grouped by decreasing fold change within one group and are colored by class: orange: Amino acids, peptides and derivatives, purple: carbohydrates and conjugates, green: lipids and lipid-like, blue: nucleosides, nucleotides and derivatives, red: organic acids, black: other. #: EOD and/or EON value below detection limit. *: and/or isomers. Data are averages ± S.E. *N* = 4–8 jars per timepoint with 3–5 plants each for **a**–**e**, *n* = 74 metabolites in total for **a**–**d**, Anova/Tukey test, *p* < 0.05 for significant differences in **e**. Details of this dataset are given in Supplementary Fig. 5.

lipids (Fig. 6). The root core metabolites did not show a diurnal signature, whereas 53% of the exudate core metabolites showed a differential abundance between exudate end-of-day and end-of night timepoints (Fig. 6).

One third of the metabolites detected in this three plant species experiment was not detected in all plant species, and thus not part of the core metabolomes. When investigating these metabolites further, half or more of the compounds was only detected in roots, but not in exudates, which is comparable to the core metabolome data (Fig. 7). Further, 25–40% of compounds were detected in roots and exudates (compared to 21% of the exudate core metabolome), and a few percent were detected in exudates only. Some metabolites were specific to one plant species. Generally, the occurrence of these 'specialized' metabolites is complex. Compounds can be found in different locations

**Root core metabolome**

3-methyl-histidine
4-acetamidobutanoic acid
4-hydroxyproline
aspartic acid
cysteic acid
ectoine
glucosaminic acid
glutathione
glycine
homoserine
N-acetyl-alanine
N-acetyl-serine *
ophthalmic acid
ornithine
3-hydroxyanthranilic acid
3-methoxytyramine *
dopamine
syringic acid
tyramine *
dihexose pk 2 (ex. maltose, cellobiose)
glucuronic acid
glyceric acid *
hexosamine
hexose phosphate
N-acetylneuraminic acid
trisaccharide (ex. raffinose)
3-methyglutaric acid
azelaic acid
glycerol phosphoric acid
maleamic acid
mevalonic acid
2'-deoxyadenosine
2'-deoxyguanosine
adenosine 5'-monophosphoric acid

cytidine 2',3'-cyclic monophosphoric acid
dAMP
guanosine 3',5'-cyclic monophosphoric acid
inosine
NAD+
nicotinamide mononucleotide
orotic acid
S-(5'-adenosyl)-homocysteine
S-(5'-adenosyl)-methionine
uric acid *
uridine 5'-diphospho-N-acetylhexosamine
uridine 5'-diphosphohexose
uridine 5'-diphosphoric acid *
uridine-5-monophosphoric acid
4-pyridoxic acid
5-hydroxyindoleacetic acid
malonic acid
pyruvic acid
tryptophan
3-dehydroshikimic acid *
4-imidazoleacetic acid
biliverdin
carnosine
choline *
lumichrome
N-acetylserotonin
putrescine
pyridoxamine
riboflavin
shikimic acid *
sphinganine

**Exudate core metabolome**

| | |
|---|---|
| 4-aminobutanoic acid | |
| 5-oxo-proline * | |
| alanine | |
| asparagine | # |
| glutamine | |
| leucine * | |
| N-trimethyllysine | # |
| pipecolic acid | |
| serine | |
| threonine * | |
| valine | |
| gluconic acid | |
| N-acetyl-hexosamine | # |
| 5-methylcytosine | # |
| 5'-methylthioadenosine | # |
| adenine | # |
| adenosine | |
| deoxycytidine | # |
| guanosine | # |
| uridine | # |
| xanthine | # |
| 4-hydroxy-2-quinolinecarboxylic acid | |
| ferulic acid | # |
| nicotinamide | |
| agmatine sulfuric acid | # |
| allantoin | # |
| glucuronolactone | |
| histamine | # |
| histidinol | |
| kynurenine | # |
| pantothenic acid | # |
| pyridoxine | # |

**Fig. 6 | Core metabolites in roots and exudates.** Metabolites detected in *A. thaliana*, *B. distachyon*, and *M. truncatula* roots or in roots as well as exudates. Metabolites are colored by class: orange: Amino acids, peptides and derivatives, purple: carbohydrates and conjugates, green: lipids and lipid-like, blue: nucleosides, nucleotides and derivatives, red: organic acids, black: other. *: and/or isomers. # changing significantly diurnally in exudates of at least one species (Anova/Tukey test, $p < 0.05$, *p*-values are given in Supplementary Data 1). Specific metabolites of this dataset are presented in Fig. 7.

(roots, exudates, or both), and in one or several of the plant species (Fig. 7).

To summarize: two thirds of compounds were detected in roots and exudates of all plant species, comprising the root and the exudate core metabolome. We speculate that the latter is of importance for interaction with the core microbiome that is associated with roots of all plant species. Further, one third of compounds was detected only in some species and locations. If future experiments confirm the exclusive presence of these compounds, they might be responsible for more specialized plant-microbe interactions.

## Discussion

In literature, exudates are collected for vastly different timespans, ranging from several hours to several days or even weeks[20,34,36–38]. With our experimental setup, a collection window of a few hours resulted in detection of most compounds. If longer collection times are necessary to increase signal strength and/or detection of specific metabolites, it should be kept in mind that some metabolite levels might have reached a plateau already. Also, it has to be taken into account that timecourse studies as the one presented here have to be interpreted cautiously, as the peak intensity of compounds is strongly influenced by presence of other ions, making the comparison between metabolite profiles collected early and late challenging.

The growth environment of *B. distachyon* further significantly influenced the exudate profile detected even when the exudates themselves were collected in the same medium. Sugar-supplemented growth medium resulted in mixotrophic *B. distachyon* growth, in increased carbohydrate exudation, and in a trend for higher amino acid exudation. This shift in exudate profile is especially important for researchers working with Arabidopsis, as they tend to be cultivated in sucrose-supplemented conditions[7,20,24]. Exudation also differed between sterile and nonsterile conditions, with a trend for higher amino acid exudation, an observation made before[28,39]. Aside from changes in amino acid levels, about half of the compounds differed in abundance between sterile and nonsterile conditions. Differential exudation in nonsterile conditions could be a result of microbial metabolism, or of altered root exudation, for example caused by microbial signals or the altered root morphology observed. In literature, distinct exudation profiles were reported for distinct exudate collection media used on plants grown in the same environment. Most striking differences are usually seen between exudates collected in deionized water compared to an equimolar solution, as used here. Generally, exudates collected in deionized show in higher metabolite levels compared with buffered solutions[15,26]. On the one hand, this could be explained with matrix effects on the metabolomics analysis, and on the other hand with biological factors such as an increase in diffusion gradient, osmotic shock, or plant cell lysis in water. When working with soil-grown plants, water:solvent mixtures were tested for exudate collection in situ[40].

Compounds detected in exudates are typically synthesized in roots. Often, exudate metabolic profiles were found to differ from root metabolic profiles[39]. In the case of rice, 63–85% of root metabolites were found in exudates[41], whereas here, about one third of root metabolites were also detected in exudates.

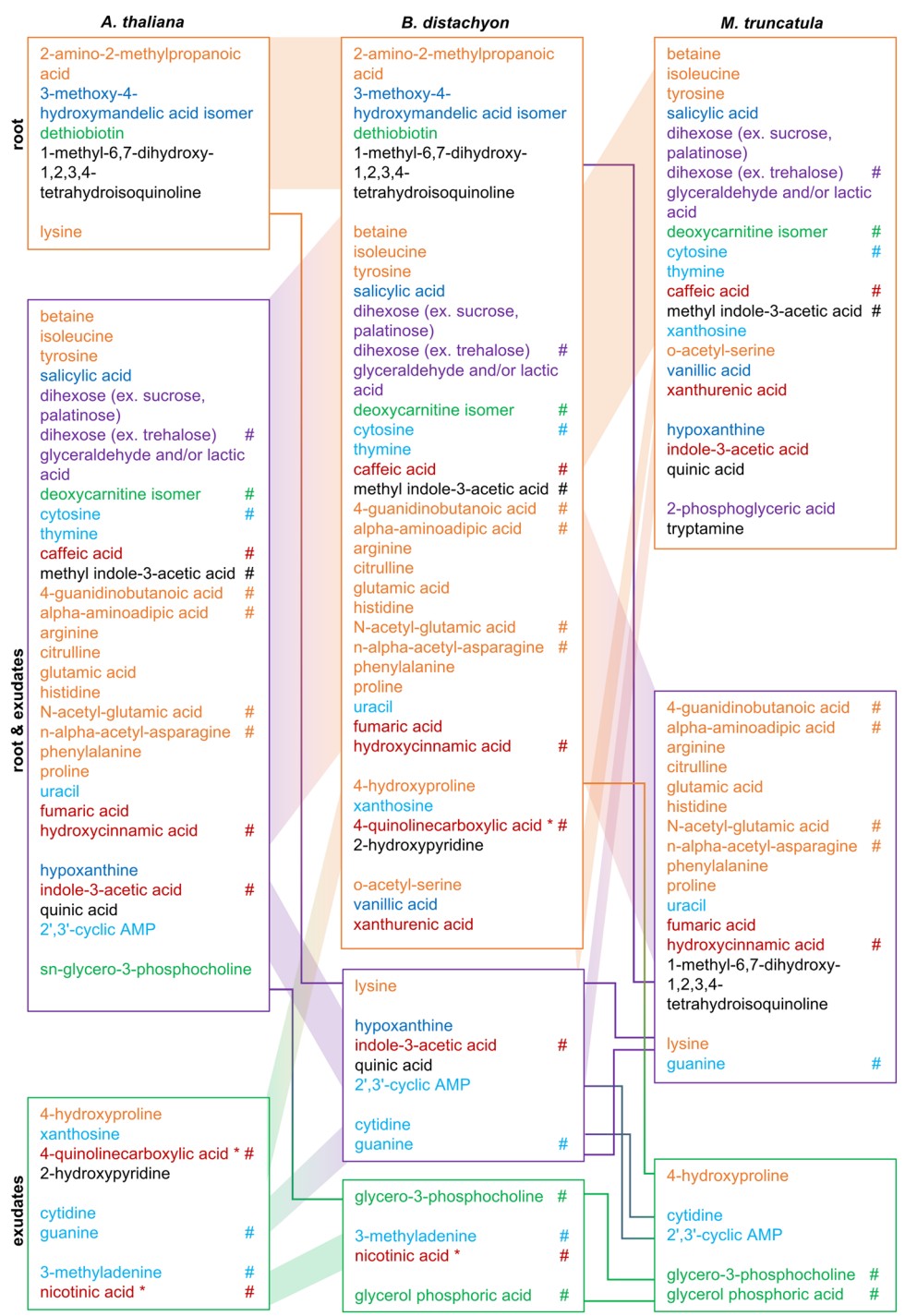

**Fig. 7 | Specific metabolites in roots and exudates.** Metabolites detected in roots and/or exudates of one or several species. A specific metabolite is present in roots (orange box), roots and exudates (purple box), or exudates (green box). Its presence in *A. thaliana*, *B. distachyon*, *M. truncatula* is indicated by the connecting colored lines. Number of jars with 3–5 plants: 3–8 for each timepoint. One representative experiment out of three total is displayed. Metabolites are colored by class: orange: Amino acids, peptides and derivatives, purple: carbohydrates and conjugates, green: lipids and lipid-like, blue: nucleosides, nucleotides and derivatives, red: organic acids, black: other. *: and/or isomers. # changing significantly diurnally (Anova/Tukey test, $p < 0.05$, $p$-values are given in Supplementary Data 1). Common metabolites of this dataset are presented in Fig. 6.

Tissue as well as exudate profiles were clearly distinct between the species, an observation made before for different species, cultivars, and ecotypes, as well as for different photosynthetic types[7,10–12,26,28,39,42,43]. Interestingly, two thirds of compounds were present in roots and exudates of *A. thaliana*, *B. distachyon* and *M. truncatula*, comprising a core metabolome of roots and exudates, respectively. We suggest to expand the idea of a core metabolome in later studies by investigating exudate profiles of additional species, and with methodologies differing from the one applied here. The description of a core set of compounds present in roots and exudates might facilitate the study of plant-microbe interactions. Roots associate with core epiphytes and endophytes, and this interaction is likely partially governed by nutrients and signaling compounds present. In addition, whereas the core metabolome might be crucial for association with the core microbiome, species-specific exudates likely result in association with specific microbes. It is well-known that various

microbes have distinct substrate preferences, and that these preferences are dynamic depending on substrate availability and presence of other microbes[4,44–46]. Thus, the presence of specific compounds in exudates or tissues likely govern microbe-microbe interactions such as competition or cross-feeding. When the metabolic profile changes with e.g. plant developmental stage, biotic or abiotic stresses, the microbial community reacts[47].

The function of specific exuded metabolites could be studied with reductionistic approaches. In analogy to the use of synthetic communities (Syncoms) as representatives of natural microbiomes, simplified exudate mixtures could be made. Single metabolites could be added or removed, and their quantity changed. The response of a microbial community to these synthetic exudates could then be studied[48]. Also, synthetic exudates could be applied to microbial communities in natural soils, mimicking the presence of plants. Exometabolomic methods could then be used to determine the substrate preferences of rhizosphere microbes[4,49], and could be used to determine which microbes prefer which exudates. A number of compounds have already been reported to influence composition of microbial communities. Among these are generalist compounds such as phenolic organic acids[4] that are detected in exudates of many plant species, and specialist compounds such as and triterpenes and benzoxazinoids exuded by Arabidopsis and grasses, respectively[25,50], which might be responsible for more specialized plant-microbe interactions.

An unexpected finding was the detection of a diurnal signature for exudates, but not for plant tissues. In literature, exudates were not always found to fluctuate diurnally. Arabidopsis grown for 3 weeks in sucrose-supplemented conditions only showed 7 out of 390 features changing diurnally[24]. Another studied showed 25% fluctuating metabolites[51], and an Arabidopsis carbon metabolism mutant exhibited changes in root respiration, as well as altered sugar, amino acid, and organic acid dynamics in export from leaves, and in roots[52,53]. We found that most compounds were of lower abundance at the end-of-night than at the end-of-day. This might reflect a lower metabolite flow from leaves to roots to the rhizosphere due to absence of carbon fixation at night, maybe coupled with a nocturnal increase in water uptake. Or it might be the consequence of a regulatory process, possibly driven by diurnally regulated genes. Microbiome abundance and transcriptional activity change diurnally, and plant clock genes were shown to be involved in this process. In addition, rhizosphere soil organic matter differed between day and night[3,13,51,54]. It is to date unclear if the reduction in exudation is the cause for changes in microbiome activity and in soil organic matter observed in the aforementioned studies. Future studies could investigate the link between carbon fixation via photosynthesis, carbon transport within the plant and into the rhizosphere, depending on additional factors such as transpiration or microbial activity.

There are quite some limitations with the studies presented here. Among them are technical challenges to the metabolomic analysis of exudates and tissues that are worth mentioning. Specifically, this study focuses on the analysis of polar metabolites using normal phase chromatography. It is well known that compounds co-eluting with metabolites can affect their ionization efficiency. In this study, we used internal standards to partially control for these effects. Further, we normalized exudates by root weight, a common practice besides for example normalizing by total carbon exuded[4]. For future work, exudate profiles of hydroponically grown plants as the ones from this study should be compared to ones grown in natural environments (different soils, soil extracts) to investigate the mode of growth (auxotrophic, mixotrophic), and the difference in metabolic profiles. Plants import a significant number of compounds from their environment, if they are able to outcompete other organisms[32,55]. Studies investigating exudation in natural systems are complicated by microbial metabolism, and soil physiochemistry. Also, plant metabolism can change

significantly with the interaction with beneficial or pathogenic organisms, such as mycorrhizal fungi or Ralstonia. Most of these quantitative and qualitative shifts in metabolic profiles are not well studied, but are central to determine the molecular mechanisms of plant-microbe interactions. First steps in this direction are for example microbial growth assays in previously collected exudates[4,45], and treatment of plants with alive or inactivated microbes to determine changes in exudation. Further, interactions of exudates with soil particles are crucial to determine for natural environments[32].

We conclude that growth and sampling conditions are a crucial determinant for the exudation profiles observed. Exudates are dynamic, varying diurnally, being distinct for the plant species investigated here. Still, many common metabolites were detected, which resulted in a first description of a root and exudate core metabolome.

## Methods
### Plant growth conditions
*Brachypodium distachyon* Bd21-3 seeds were dehusked and sterilized in 70% v/v ethanol for 30 s, and in 6% v/v NaOCl, 0.1% v/v Triton X-100 for 5 min, followed by five wash steps in water. Seedlings were germinated on 0.5× Murashige & Skoog plates (0.5× MS, MSP01, Caisson Laboratories, USA; 6% w/v Bioworld Phytoagar, 401000721, Fisher Scientific, USA; pH 5.7) in a 16 h light/8 h dark regime at 24 °C for three days. *Medicago truncatula* cv Jemalong seeds were sterilized by a 30 min incubation in 70% v/v ethanol, 4 rinses in sterile water, 30 min incubation in 6% v/v NaOCl, and 4 rinses in sterile water. Seeds were imbibed by a 4 h incubation in sterile water, followed by germination on 0.5× MS plates for three days. *Arabidopsis thaliana* Col-0 seeds were sterilized by immersion in 70% v/v ethanol for 15 min, followed by immersion in 100% v/v ethanol for 15 min, and drying. Seeds were germinated for 14 days on 0.5x MS plates, to avoid drowning of the plants in jars.

Weck jars (743, Glashaus Inc., USA) were rinsed five times with MilliQ water, sprayed with 70% v/v ethanol, treated with UV for 1 h in a laminar flow hood, and dried over night. The jars were filled with 150 ml 5 mm glass beads serving as inert, solid substrate to hold the plants upright, and with 50 ml 0.5× MS (MSP01, Caisson Laboratories, USA). For 0.5MS + suc conditions, the medium was supplemented with 2% w/v sucrose (S0389, Sigma-Aldrich, USA). Experimental control jars were set up exactly the same, but they did not contain plants. The control jars were treated the same as the plant-containing jars throughout the experiment.

Three seedlings of *B. distachyon* and *M. truncatula* and four seedlings of *A. thaliana* were transferred into each jar. To enable gas exchange in a sterile environment, two strips of micropore tape (56222-182, VWR) were placed across the jar opening, and the lid was wrapped with micropore tape (56222-110, VWR)(Supplementary Fig. 1). The nonsterile plants were set up the same way, except the last micropore tape, to enable gas exchange with the environment. Plants were grown to 21 days after germination in a 16 h light/8 h dark regime at 24 °C with 150 μmol m$^{-2}$ s$^{-1}$ illumination for three weeks. The medium was replaced weekly, and the day before collecting root exudates. Sterility of the jars was tested before exudate collection by plating 50 μl medium on Luria-Bertani (LB) plates, following by three days incubation at 24 °C.

### Root exudate collection
The medium in jars was replaced with 50 ml 0.5× MS medium. Plants were incubated for two hours or the specified amount of time at 24 °C with 150 μmol m$^{-2}$ s$^{-1}$ illumination. Exudates were collected mid-day, except end-of-day samples, which were collected for the last 2 h of the day, and the end-of-night samples, which were collected for the last 2 h of the night. Root exudates of plants grown in sucrose-supplemented medium were also collected in sucrose-free 0.5× MS to avoid ion suppression in downstream LC/MS analysis. The exudates were

collected by pipetting, filtered through a 0.45 μm filter (4654, PALL Life Sciences, USA), frozen at −80 °C, and lyophilized (Labconco FreeZone lyophilizer).

## Plant tissue collection

For the diurnal experiment, root and shoot tissue of *A. thaliana*, *B. distachyon*, and *M. truncatula* was sampled at end-of-day and end-of-night time points, and immediately frozen on dry ice. Tissue fresh weight was recorded with pre-weighted tubes, and samples were stored at −80 °C until further processing.

## Liquid chromatography sample preparation

Root and shoot tissue samples were ground to powder with a bead mill for two times 1.5 min, 30 s$^{-1}$ frequency. The frozen samples were transferred to ice, 1 ml 4 °C LC/MS grade methanol was added, and samples were vortexed to suspend the material. Extraction controls were prepared by adding 1 ml of LC/MS grade methanol to empty tubes. The controls were treated the same way as the samples. The tubes were incubated on ice for 1 h with frequent vortexing. Samples were centrifuged for 5 min at 10,000 g and 4 °C. The supernatant was transferred to a new tube, and the methanol was evaporated at 24 °C under vacuum until dry. The extraction with 1 ml methanol on ice was repeated a second time, and the supernatants of the samples were pooled.

Root exudate samples were resuspended in 3 ml LC/MS grade methanol (CAS 67-56-1, Honeywell Burdick & Jackson, Morristown, NJ, USA), vortexed three times for 10 s, sonicated for 20 min in a water bath at 24 °C. For salt precipitation, samples were incubated at 4 °C for 16 h, centrifuged for 5 min at 5000 g and 4 °C, supernatants were transferred to new microcentrifuge tubes, and evaporated at 24 °C under vacuum until dry. The samples were resuspended in 500 μl LC/MS grade methanol, and a second salt precipitation step was performed as described above.

Finally, exudate and tissue samples were resuspended in 100% LC/MS grade methanol with 15 μM internal standards (767964, Sigma-Aldrich, USA), with a volume relative to tissue fresh weight. For exudates, the root fresh weight was used.

## Metabolomics analysis

Metabolites were chromatographically separated using hydrophilic liquid interaction chromatography and detected with a Q Exactive Hybrid Quadrupole-Orbitrap Mass Spectrometer equipped with a HESI-II source probe (ThermoFisher Scientific). For chromatographic separations, an Agilent InfinityLab Poroshell 120 HILIC-Z column (2.1 × 150 mm, 2.7 μm) was used on an Agilent 1290 series HPLC system. Separation and detection parameters are defined in Supplementary Data 2. Internal and external standards were included for quality control purposes, with blank injections between every unique sample. LCMS data were compared with an in-house library of reference standards using Metabolite Atlas (https://github.com/biorack/metatlas). Metabolite identifications were made based on retention time (within 1 min vs. standard), fragmentation spectra (manual inspection), and accurate mass (within 20 ppm). LC/MS data quality was ascertained by analyzing quality control samples that were included at the beginning, during, and at the end of the run. Internal standards were used to assess sample-to-sample consistency for peak area and retention times.

Metabolites were excluded from analysis if the mean intensity of the experimental treatments was the same or lower compared to the mean intensity of the experimental blanks. For the time-course experiment, samples were in addition compared to the 0 min control time point. Further, metabolite intensities were expressed as percentage of the maximum peak height, allowing for relative comparison of peak heights between samples (e.g. if a compound of interest is significantly different between samples), but not for absolute metabolite

level quantification (e.g. μg of a compound of interest per gram tissue). Chemical classes were assigned to metabolites with the ClassyFire compound classification system[56].

## Statistical analysis

To explore the variation between experimental conditions, the metabolite profiles were PCA-ordinated, and the 95% confidence level was displayed as ellipses for each treatment. Hierarchical clustering analysis with a Bray Curtis Dissimilarity Matrix was performed with the python 2.7 Seaborn package. Metabolite significance levels were analyzed with the python SciPy ANOVA test coupled to a python Tukey's honestly significant difference test with alpha = 0.05 corresponding to a 95% confidence level.

## Reporting summary

Further information on research design is available in the Nature Portfolio Reporting Summary linked to this article.

## Data availability

Source data are provided with this paper. The metabolomic raw data files are stored as a MassIVE dataset MSV000090869, https://doi.org/10.25345/C5GB1XN3T. Source data are provided with this paper.

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

## Acknowledgements

We thank Prof. Sharon Long, Stanford, USA for providing *Medicago truncatula* seeds, and Katherine Whiting for supporting the root morphology analysis. J.S. was supported by a NSF grant to University of California Berkeley (NSF Proposal 1617020). J.S. and S.M. are supported by a grant to J.S. of the Swiss National Science Foundation (SNSF, Proposal PROOP3_185831). K.Z., S.K., and T.R.N. are supported by the m-CAFEs Microbial Community Analysis & Functional Evaluation in Soils, (m-CAFEs@lbl.gov) a Science Focus Area led by Lawrence Berkeley National Laboratory and the work was performed at the U.S. Department of Energy Joint Genome Institute, a DOE Office of Science User Facility both of which are supported by the U.S. Department of Energy, Office of Science, Office of Biological & Environmental Research under contract number DE-AC02-05CH11231.

## Author contributions

J.S. and T.R.N. conceived the project, JS designed the experiments, J.S., S.M., S.K., and K.Z. conducted the experiments, J.S. analyzed the data and wrote the manuscript with input from all authors.

## Competing interests

The authors declare no competing interests.
