## [Peer Review File · Nature Communications]

The core metabolome and root exudation dynamics of three phylogenetically distinct plant speciesReviewer #1 (Remarks to the Author):

Key results

The central message of this paper is that the three key model species used in plant sciences have both distinct and common metabolomes in their leaf and root organs, and in their root exudates. Further the root exudate metabolomes vary with length and time of day of sampling, in a species dependent manner. This paper is significant in that it demonstrates the suite of metabolic compounds that plants control in space and time. The information provides a strong foundation for further mechanistic studies that address the processes regulating these metabolomes. By far the emphasis is on the root exudates, an understudied metabolome that regulates plant-mediated soil processes. The authors grow the plants in glass beads and nutrient solution with and without microorganisms and with and without sugar. The conditions are common to many molecular plant laboratories. The methods and results are relevant to the reductionist, simplified research conditions that are used by many researchers and are essential as a starting point to discover how plants regulate their soil environments.

Data, methodology and interpretation

The data is of a high quality. The authors check for sterility using a plating technique. Although it is possible that organisms were present that were not detected in the plating, the clarity of the methods allow that interpretation. I did not find any flaws in the data. To my knowledge, this is the most comprehensive quantitative coverage of the metabolome of shoot, root and root exudate fractions of plants, in the sense that the analytical techniques have covered the largest suite of metabolites possible in each sample on a per plant basis. Many other studies measure, for example, the root exudates and not the root and shoot tissues; or measure only products of a particular pathways such organic acids.

The data interpretation within the manuscript is largely around the diversity and commonality of compounds measured in the metabolomes, across species and sampling. This is robust because the authors are careful to place the results within the laboratory, simple conditions of the experiment; e.g., they do not extend them to soil, complex climatic conditions of nature where numerous studies have collected root exudates and found variation related to plant age, time of day, season and soil types.

Clarity

The methods for these challenging experiments are presented clearly in a high level of detail and all data is provided in the supplemental files, making the experiments easy to test for repeatability. The paper was a pleasure to read. The metabolomics methods of the paper are highly valuable to the research community.

Conclusions

The central conclusion of the paper, beyond the value of the methods presented, appears to be that plants have a "core metabolome". This conclusion aligns with a recent emphasis on finding a "core microbiome" for species. The extensive metabolomics of shoot, root and exudates across three model species, representing the grass family, brassicaceae family and legume family, supports the conclusion that there is core metabolome among these important model species. Additionally, it supports the knowledge that plants share common primary and secondary metabolic pathways. However, I feel this conclusion could be tempered to highlight the many species not covered (eg, perennials versus the young, annuals here). Additionally, they could add the word "root" to exudates, or "rhizosphere" to the exudates since no shoot exudates were measured and it is very possible that shoots also release compounds.

Significance

In my view, this paper is significant for its methods and presenting very complete metabolomes of roots, shoots and root exudates of three important model plants used in molecular biology today. The fact that diurnal patterns and effects of sterility and sugar in the growing media are presented, will emphasise anew the many questions outstanding about the mechanisms regulating diversities. The published literature has reports of most of the findings of this paper; however not

as comprehensively or systematically. This paper lays the foundation for conceptually considering the core metabolome of plants, and how this changes in space and time.

Analytical approach

The analytical approaches are simple and appropriate. All data is presented.

Suggested improvements

Major.

The paper would be greatly enhanced with experimental treatments to explore the diurnal pattern of some exudates. Diurnal changes in exudates has been demonstrated in the literature, leaving the question about what is regulating them. A relatively straightforward experiment using the current set up, would be to test the role of transpiration and photosynthesis by imposing a dark period in the day/ light period during the night. Similarly, older literature using ¹⁴C labelling of shoots, showed an increase in carbon release from roots in non-sterile conditions. This important phenomenon has not been explored mechanistically. The current set up could be used to challenge plants in experiments with a specific suite or singular microorganisms to see how exudates respond in time.

Minor.

Make clear in Fig 1 and Fig 2 legends (all ideally), how many plants from how many jars from how many repeat experiments are presented. Sometimes the legends mention the number of plants averaged (eg., 7 to 12) but is not clear if these are pseudo replicates within a jar or replicates across independent experimental runs.

Consider moving Supplemental Figure S8 to the main text. This was a particularly useful way to visualise the common and unique compounds to the root and root exudate metabolomes.

Consider removing the root architecture data for *Brachypodium* in its current form in S4. This data seemed out of place because it is not provided for *Arabidopsis* or *Medicago*, and is not explored mechanistically. The response of root development to the presence or absence of microorganisms is reported frequently. In its current form and with these limited experiments, this data raises more questions than answers within the paper.

References

References are appropriate and used well to introduce the paper and to discuss the results. I highlighted above that there are reports of specific exudates following diurnal patterns in the literature, as well as older papers finding that sterility/non-sterility can alter exudate release. Although challenging to find, these would be good to include.

Reviewer #2 (Remarks to the Author):

My comments and the suggested areas of improvement for this manuscript lie in three parts.

First, while the authors have done a good job investigating several technique factors (e.g., duration time, growth media) as well as plant species that affect exudate profiles, the novelty of this study is not immediately clear. It is well known that root exudates are sensitive to the choice of exudation sampling techniques. As the authors stated in the manuscript, it has been well documented that exudate profiles vary across species (L 320-322) and can be affected by growth media (L301-303).

It is also known that the duration time influences the collected exudates, which are partly due to metabolite re-uptake (L133-135, also see Warren, 2015 <https://doi.org/10.1007/s11104-015-2612-4>). In addition, in this manuscript, the technique recommendation of 2h or 4h for collection time seems a bit arbitrary and overgeneralizing. The three species exhibited highly species-specific

exudation dynamics (Fig. 1). While *M. truncatula* did not change much over time, *B. distachyon* continuously shifted from 0.5h to 4h. For the dynamics of specific compounds, only *B. distachyon* was reported (Fig. S3), which showed half of the compounds continuously increased over time while the other half decreased or remained the same between 1d and 4d (however, the authors stated, "intensities decreased mostly between 1d and 4d", which is not consistent with Fig. S3). How this highly species-specific and compound-specific dynamic pattern is linked to a proposed optimum collection time of 2-4h is unclear.

Identifying specific knowledge gaps and how this dataset would help fill the gaps will help improve this manuscript.

Secondly, one novelty of this manuscript may lie on the "metabolome core" they observed and proposed as potentially crucial for plant-microbe interactions. However, there has been relatively little effort to discuss the biological meaning/significance of this concept. As for now, the manuscript reads like a method paper, while the method part is not necessarily novel. How are the specific compounds in this "core" may be crucial for plant-microbe interactions? How may this "core" concept reshape the field's understanding of plant-microbe interactions? Discussing these points could boost the significance of this study.

Finally, some of the results are not well supported by the data.

- 1) The recommendation of 2h-4h optimum collection time seems arbitrary. Please see above.
- 2) The difference in chemical composition between sterile and non-sterile conditions may not necessarily arise from different growth mediums. Seedlings with non-sterile conditions exhibited considerably higher growth than those with sterile conditions (Fig. 2). Developmental stages/growth status may also affect exudation, rather than growth medium. Moreover, the difference in chemical composition may be due to the microbial metabolites in non-sterile conditions.
- 3) Fig. 5. The statistical analysis for the diurnal changes is unclear. Fig. 5A, the separation was not statistically tested. *M. truncatula* may not have significant diurnal signal as this species had small sample size ($n=3$) and its day and night exudate profiles were not well separated. Fig. 5E. Where are the SD bars? Some of the changes were very small. Were they significant? SD bars and details of tests for significant differences are needed for data interpretation.

Other comments.

L331-336. This study and the "metabolome core" were focused on primary metabolites. I would recommend focusing on primary metabolites rather than expanding discussion on specific specialized compounds.

Fig. 5. This figure is confusing. Do you mean only the metabolites that increased in day vs. night in *A. thaliana* were listed on the top graph? But for uridine, both *A. thaliana* and *B. distachyon* increased.

REVIEWER COMMENTS

Reviewer #1 (Remarks to the Author):

Key results

The central message of this paper is that the three key model species used in plant sciences have both distinct and common metabolomes in their leaf and root organs, and in their root exudates. Further the root exudate metabolomes vary with length and time of day of sampling, in a species dependent manner. This paper is significant in that it demonstrates the suite of metabolic compounds that plants control in space and time. The information provides a strong foundation for further mechanistic studies that address the processes regulating these metabolomes. By far the emphasis is on the root exudates, an understudied metabolome that regulates plant-mediated soil processes. The authors grow the plants in glass beads and nutrient solution with and without microorganisms and with and without sugar. The conditions are common to many molecular plant laboratories. The methods and results are relevant to the reductionist, simplified research conditions that are used by many researchers and are essential as a starting point to discover how plants regulate their soil environments.

Data, methodology and interpretation

The data is of a high quality. The authors check for sterility using a plating technique. Although it is possible that organisms were present that were not detected in the plating, the clarity of the methods allow that interpretation. I did not find any flaws in the data. To my knowledge, this is the most comprehensive quantitative coverage of the metabolome of shoot, root and root exudate fractions of plants, in the sense that the analytical techniques have covered the largest suite of metabolites possible in each sample on a per plant basis. Many other studies measure, for example, the root exudates and not the root and shoot tissues; or measure only products of a particular pathways such organic acids.

The data interpretation within the manuscript is largely around the diversity and commonality of compounds measured in the metabolomes, across species and sampling. This is robust because the authors are careful to place the results within the laboratory, simple conditions of the experiment; e.g., they do not extend them to soil, complex climatic conditions of nature where numerous studies have collected root exudates and found variation related to plant age, time of day, season and soil types.

Clarity

The methods for these challenging experiments are presented clearly in a high level of detail and all data is provided in the supplemental files, making the experiments easy to test for repeatability. The paper was a pleasure to read. The metabolomics methods of the paper are highly valuable to the research community.

Conclusions

The central conclusion of the paper, beyond the value of the methods presented, appears to be that plants have a “core metabolome”. This conclusion aligns with a recent emphasis on finding a “core microbiome” for species. The extensive metabolomics of shoot, root and exudates across three model species, representing the grass family, brassicaceae family and legume family, supports the conclusion that there is core metabolome among these important

model species. Additionally, it supports the knowledge that plants share common primary and secondary metabolic pathways. However, I feel this conclusion could be tempered to highlight the many species not covered (eg, perennials versus the young, annuals here). Additionally, they could add the word “root” to exudates, or “rhizosphere” to the exudates since no shoot exudates were measured and it is very possible that shoots also release compounds.

Significance

In my view, this paper is significant for its methods and presenting very complete metabolomes of roots, shoots and root exudates of three important model plants used in molecular biology today. The fact that diurnal patterns and effects of sterility and sugar in the growing media are presented, will emphasise anew the many questions outstanding about the mechanisms regulating diversities. The published literature has reports of most of the findings of this paper; however not as comprehensively or systematically. This paper lays the foundation for conceptually considering the core metabolome of plants, and how this changes in space and time.

Analytical approach

The analytical approaches are simple and appropriate. All data is presented.

Dear reviewer, thank you for the detailed summary, and for the appreciation of the work. Of course you are correct in that the work focuses on root exudates (not analyzing leaf exudates). We added ‘root exudates’ to the title as well as to the abstract to make the focus on belowground exudation clear.

We agree of course that the concept of a ‘core metabolome’ is still new and the list of core compounds identified in this study represents a starting point towards defining a larger concept of ‘core microbiome’. We are looking forward to additional work also taking into account plant species not covered here, as mentioned by the reviewer.

Suggested improvements

Major.

The paper would be greatly enhanced with experimental treatments to explore the diurnal pattern of some exudates. Diurnal changes in exudates has been demonstrated in the literature, leaving the question about what is regulating them. A relatively straightforward experiment using the current set up, would be to test the role of transpiration and photosynthesis by imposing a dark period in the day/ light period during the night. Similarly, older literature using ¹⁴C labelling of shoots, showed an increase in carbon release from roots in non-sterile conditions. This important phenomenon has not been explored mechanistically. The current set up could be used to challenge plants in experiments with a specific suite or singular microorganisms to see how exudates respond in time.

Thank for these thoughts. We fully agree that the diurnal pattern of exudation is a very interesting (and understudied) field of research. The proposed experimental outline investigating photosynthesis rate, carbon labeling and co-cultivation with microbes sounds very exciting. We think that this would be an excellent follow-up study, likely with considerable impact. In our eyes however, the proposed experiments are beyond the scope of the current work. We added the following sentence to the discussion section: “Future studies could investigate the link between carbon fixation via photosynthesis, carbon transport within

the plant and into the rhizosphere, depending on additional factors such as transpiration or microbial activity.”

We also added additional thoughts to the discussion on how exudate profiles are influenced aside from diurnal considerations: “Also, plant metabolism can change significantly with the interaction with beneficial or pathogenic organisms, such as mycorrhizal fungi or *Ralstonia*. Most of these quantitative and qualitative shifts in metabolic profiles are not well studied, but are central to determine the molecular mechanisms of plant-microbe interactions. First steps in this direction are for example microbial growth assays in previously collected exudates 3,46, and treatment of plants with alive or inactivated microbes to determine changes in exudation. Further, interactions of exudates with soil particles are crucial to determine for natural environments 33.”

Minor.

Make clear in Fig 1 and Fig 2 legends (all ideally), how many plants from how many jars from how many repeat experiments are presented. Sometimes the legends mention the number of plants averaged (eg., 7 to 12) but is not clear if these are pseudo replicates within a jar or replicates across independent experimental runs.

Thank you for pointing this out. Generally, multiple plants are grown within one jar to produce a high amount of exudates, and also to allow for some averaging of exudate profiles across plants. A minimum of three jars per conditions are set up from which exudates are sampled (data points displayed). There are no technical replicates, only multiple sampling from the same jars in case of e.g. the diurnal setup or the timecourse. We added the information to all figure legends where applicable, such as Fig 1: “Number of jars with 3-5 plants: 3-4 for each timepoint. One representative experiment out of three total is displayed.”

Consider moving Supplemental Figure S8 to the main text. This was a particularly useful way to visualise the common and unique compounds to the root and root exudate metabolomes.

Thank you for the suggestion, we moved Figure S8 to the main text as Figure 7.

Consider removing the root architecture data for *Brachypodium* in its current form in S4. This data seemed out of place because it is not provided for *Arabidopsis* or *Medicago*, and is not explored mechanistically. The response of root development to the presence or absence of microorganisms is reported frequently. In its current form and with these limited experiments, this data raises more questions than answers within the paper.

Thank you for the suggestion. We added the root morphology data here because *Brachypodium* grown in the different experimental conditions (sterile, nonsterile, sucrose-supplemented) looked quite different, as also seen by the tissue masses (Fig 2). Although this experiment was not done for all three species, we still think the morphology data might be of some interest to a variety of readers. But we agree that it raises some questions (such as the doubling in lateral root number in nonsterile conditions), which are not addressed in this manuscript, but might provide interesting starting points for follow-up studies, as mentioned in the discussion. Thus, we would like to keep the supplemental figure.

References

References are appropriate and used well to introduce the paper and to discuss the results. I highlighted above that there are reports of specific exudates following diurnal patterns in the literature, as well as older papers finding that sterility/non-sterility can alter exudate release. Although challenging to find, these would be good to include.

Thank you for the input. We expanded the introduction with the following paragraphs (new text in italic) on diurnal exudation and exudation in sterile/nonsterile conditions:

Microbiome composition and exudation changes with plant developmental stage, diurnal timepoint, and with abiotic and biotic stresses 4,13–19. *For example, sugar exudation decreases and organic acid exudation increases along a developmental gradient in several monocot and dicot plant species 14,15,20,21. Also, diurnal fluctuations were observed for some exuded compounds such as some lipids²², single organic acids such as citrate²³ and mugineic acid²⁴, and flavonoids and glucosinolates²⁵.*

(...) Exudate collection duration also influences the metabolic profile. The rate of total organic carbon exudation (carbohydrates and organic acids) was estimated higher when rice exudates were collected for 2 h than for 4 h or 6 h, independent of the plant developmental stage¹⁵. External amino acid concentrations of a variety of plant species increased in the first few hours of exudation up to one day and remained stable afterwards^{15,29}. Another major determinant for detection of exudates is the sterile or nonsterile growth environment. For tomato grown in nonsterile conditions, no sugars or organic acids could be measured in exudates, likely due to microbial activity, whereas sterile tomato exhibited organic acid exudation in the μM range within hours³⁰. In nonsterile setups, sterilizing agents can be added to block microbial metabolism, but they likely alter the plant exudation profile as well²⁸. However, the dynamics of exudation of most chemical classes remains unexplored.

To the revised version, we added the DOI Link to the raw metabolomics data. The dataset is currently still private to enable changes if need be. Here are the login details:

Login to MassIVE database:

<https://massive.ucsd.edu/ProteoSAFe/dataset.jsp?task=460de2fdaf804b21b2da4a33b2d84da4>

Username: MSV000090869_reviewer

password: test

DOI for future accessibility: <https://doi.org/doi:10.25345/C5GB1XN3T>

Dataset ID: MSV000090869

Reviewer #2 (Remarks to the Author):

My comments and the suggested areas of improvement for this manuscript lie in three parts.

First, while the authors have done a good job investigating several technique factors (e.g., duration time, growth media) as well as plant species that affect exudate profiles, the novelty of this study is not immediately clear. It is well known that root exudates are sensitive to the choice of exudation sampling techniques. As the authors stated in the manuscript, it has been well documented that exudate profiles vary across species (L 320-322) and can be affected by

growth media (L301-303).

It is also known that the duration time influences the collected exudates, which are partly due to metabolite re-uptake (L133-135, also see Warren, 2015 <https://doi.org/10.1007/s11104-015-2612-4>). In addition, in this manuscript, the technique recommendation of 2h or 4h for collection time seems a bit arbitrary and overgeneralizing. The three species exhibited highly species-specific exudation dynamics (Fig. 1). While *M. truncatula* did not change much over time, *B. distachyon* continuously shifted from 0.5h to 4h. For the dynamics of specific compounds, only *B. distachyon* was reported (Fig. S3), which showed half of the compounds continuously increased over time while the other half decreased or remained the same between 1d and 4d (however, the authors stated, “intensities decreased mostly between 1d and 4d”, which is not consistent with Fig. S3). How this highly species-specific and compound-specific dynamic pattern is linked to a proposed optimum collection time of 2-4h is unclear.

We are glad that the reviewer agrees in that exudation is affected by experimental conditions. It is also correct of course that single observations on that have been published. However, we see that in the field of plant-microbe interactions especially, these results are often neglected and the conditions for exudate sampling are often not well considered or reported. This is why we think it should be made clear that the exudate collection methodology is indeed very important. As a second thought: plants adjust root morphology and exudation to their environment, and thus, exudation profiles obtained in one (natural or artificial) environment cannot be directly compared to data obtained in a different setup. Still, this is common in the field. Also for this reason, we think that providing the methodological data in this paper is useful.

The reviewer is correct that we are not the first ones comparing exudates of different species. We added a sentence in the introduction addressing this (new sentence in italics): “Microbiome composition and exudation changes with plant developmental stage, diurnal timepoint, and with abiotic and biotic stresses 4,13–19. *For example, sugar exudation decreases and organic acid exudation increases along a developmental gradient in several monocot and dicot plant species 14,15,20,21.*”

In this manuscript however, we provide an extensive list of compounds from many chemical classes, and we separate the dataset into core and specific metabolites. In this work we introduce a concept of “core metabolome” of plants. We see this as a novel concept that is of interest particularly in conjunction with the core microbiome, and will be expanded upon by the scientific community.

Regarding the timecourse experiment: we reworded the unclear sentence “Intensities decreased mostly between 1d and 4d, if at all” to “Most compound intensities increased over time. Decreases were only observed in few instances between 1 d and 4 d.”

Regarding the 2d-4h collection window: we proposed this window based on different considerations. First, a long collection time of one to few days seems to result in higher intensities for many compounds, but other compound intensities seem to have saturated. In the second case, this might mean that a balance between exudation and reuptake was reached. The linear range for most compounds seems to start at a couple of hours collection, and to end around one day. Second, for time-resolved experiments such as a diurnal timecourse, a few hours of collection is the maximum to obtain resolved data. For the compounds depicted in Fig S3, 30 min and 1h seem to be on the lower end regarding detectability (mostly flat dynamics although 2x longer collection duration). Thus, based on the timepoints available,

we suggested 2-4 h. From a more general point of view, we could suggest a collection window of a few hours. We changed the text accordingly. E.g. L134: “For experiments with high temporal resolution, we recommend a collection window of a few hours.”

Identifying specific knowledge gaps and how this dataset would help fill the gaps will help improve this manuscript.

Thank you for the observation. We added the following paragraph to the introduction: “Although it becomes clear that exudation is a dynamic process, most parameters shaping exudation have not been systematically explored. Most studies focus on how a few experimental or biological parameters change exudation of one or few chemical classes. Comprehensive studies analyzing the dynamics of a large number of exuded metabolites systematically for the influence of experimental factors such as duration of exudate collection or impact of growth medium on exudation are missing, as well as a comprehensive analysis of similarities and differences in exudates of different plant species.”

Secondly, one novelty of this manuscript may lie on the “metabolome core” they observed and proposed as potentially crucial for plant-microbe interactions. However, there has been relatively little effort to discuss the biological meaning/significance of this concept. As for now, the manuscript reads like a method paper, while the method part is not necessarily novel. How are the specific compounds in this “core” may be crucial for plant-microbe interactions? How may this “core” concept reshape the field’s understanding of plant-microbe interactions? Discussing these points could boost the significance of this study.

Thank you for the suggestion to expand on the more metabolome idea. We added the following paragraph to the discussion (new part in italics): Roots associate with core epiphytes and endophytes, and this interaction is likely partially governed by nutrients and signaling compounds present. In addition, whereas the core metabolome might be crucial for association with the core microbiome, species-specific exudates likely result in association with specific microbes. *It is well-known that various microbes have distinct substrate preferences, and that these preferences are dynamic depending on substrate availability and presence of other microbes^{3,40-42}. Thus, the presence of specific compounds in exudates or tissues likely govern microbe-microbe interactions such as competition or cross-feeding. When the metabolic profile changes with e.g. plant developmental stage, biotic or abiotic stresses, the microbial community reacts⁴³.*

In analogy to reductionistic approaches widely utilizes as the application of synthetic communities (Syncoms) in plant microbiome studies, the function of exudates could be studied. If the (exudate) core metabolome of plants is defined qualitatively and quantitatively, synthetic exudate mixtures can be made allowing to study the role of specific compounds on microbes by changing their abundance⁴⁴. Synthetic exudates could be applied to Syncoms, or to microbial communities in natural soils, mimicking the presence of plants. Exometabolomic methods could then be used to determine the substrate preferences of rhizosphere microbes^{3,45}. A number of compounds have already been reported to influence composition of microbial communities. Among these are generalist compounds such as phenolic organic acids³ that are detected in exudates of many plant species, and specialist compounds such as and triterpenes and benzoxazinoids exuded by Arabidopsis and grasses, respectively^{20,46}, which might be responsible for more specialized plant-microbe interactions.

Finally, some of the results are not well supported by the data.

1) The recommendation of 2h-4h optimum collection time seems arbitrary. Please see above.

Thank you, we updated the manuscript to “a few hours”, see also comment above.

2) The difference in chemical composition between sterile and non-sterile conditions may not necessarily arise from different growth mediums. Seedlings with non-sterile conditions exhibited considerably higher growth than those with sterile conditions (Fig. 2). Developmental stages/growth status may also affect exudation, rather than growth medium. Moreover, the difference in chemical composition may be due to the microbial metabolites in non-sterile conditions.

The reviewer is correct that in nonsterile conditions, some of the metabolic changes might be due to the microbes present rather than by the plant. This thought is already mentioned in the respective results section: “Thus, the increase in metabolite levels in nonsterile conditions might be due to an alteration in plant metabolism, maybe in response to microbial presence, or due to the additional presence of microbial metabolism.”

3) Fig. 5. The statistical analysis for the diurnal changes is unclear. Fig. 5A, the separation was not statistically tested. *M. truncatula* may not have significant diurnal signal as this species had small sample size (n=3) and its day and night exudate profiles were not well separated. Fig. 5E. Where are the SD bars? Some of the changes were very small. Were they significant? SD bars and details of tests for significant differences are needed for data interpretation.

The significantly different metabolites are given as percent in Fig. 5D (e.g. 32% of *Arabidopsis* exuded metabolites were significant between end of day and end of night). We agree that in *Medicago*, the low number of significant metabolites might be due to the low number of replicates. This was caused by technical difficulties in obtaining sterile jars for *Medicago*. Also, the *Medicago* seeds were the ones with most genetic diversity (as judging from observed phenotypes).

For Fig. 5E, error bars were added. The end of day – end of night significant differences are given by the figure layout: in the top graph, metabolite levels were only significantly different for *Arabidopsis* (as labeled on the graph). The lower graph displays the metabolites that were significantly different for the other species or a combination of species.

Other comments.

L331-336. This study and the “metabolome core” were focused on primary metabolites. I would recommend focusing on primary metabolites rather than expanding discussion on specific specialized compounds.

Thank you for the suggestion. We mentioned the exuded secondary metabolites (benoxazinoids, triterpenes) for two reasons: 1. For the core vs. specialized exudation and

microbiome mentioned just above and 2. As there are not many examples published yet of primary metabolites altering microbiome structure. Thus, we would like to keep this section.

Fig. 5. This figure is confusing. Do you mean only the metabolites that increased in day vs. night in *A. thaliana* were listed on the top graph? But for uridine, both *A. thaliana* and *B. distachyon* increased.

We are sorry for the confusion. Indeed, the reviewer reads the data correctly in that on the first graph, only the *A. thaliana* end of day – end of night differences are significant. For uridine, the fold change in *B. distachyon* is not significant. We hope that addition of the error bars helps with understanding.

To the revised version, we added the DOI Link to the raw metabolomics data. The dataset is currently still private to enable changes if need be. Here are the login details:

Login to MassIVE database:

<https://massive.ucsd.edu/ProteoSAFe/dataset.jsp?task=460de2fdaf804b21b2da4a33b2d84da4>

Username: MSV000090869_reviewer

password: test

DOI for future accessibility: <https://doi.org/doi:10.25345/C5GB1XN3T>

Dataset ID: MSV000090869

Reviewer #1 (Remarks to the Author):

The comments have been addressed well, and were all taken on board.